# A Comprehensive Review Uncovering the Challenges and Advancements in the In Vitro Propagation of *Eucalyptus* Plantations

**DOI:** 10.3390/plants12173018

**Published:** 2023-08-22

**Authors:** Vikas Sharma, Arun Karnwal, Shivika Sharma, Barkha Kamal, Vikash S. Jadon, Sanjay Gupta, Iyyakkannu Sivanasen

**Affiliations:** 1School of Bioengineering and Bioscience, Lovely Professional University, Phagwara 144411, Punjab, India; vikas.25269@lpu.co.in (V.S.); anki2026as@gmail.com (A.); arun.20599@lpu.co.in (A.K.); shivikasharma25@gmail.com (S.S.); 2DBS (PG) College, Dehradun 248001, Uttarakhand, India; barkhabiotech2005@gmail.com; 3Himalayan School of Biosciences, Swami Rama Himalayan University, Jolly Grant Dehradun 248016, Uttarakhand, India; vsjadon@srhu.edu.in (V.S.J.); sanjaygupta@srhu.edu.in (S.G.); 4Department of Bioresource and Food Science, Institute of Natural Science and Agriculture, Konkuk University, Seoul 05029, Republic of Korea

**Keywords:** *Eucalyptus*, tissue culture, F1 hybrids, clonal propagation, micropropagation, germplasm

## Abstract

The genus *Eucalyptus* is a globally captivated source of hardwood and is well known for its medicinal uses. The hybrid and wild species of *Eucalyptus* are widely used as exotic plantations due to their renowned potential of adapting to various systems and sites, and rapid large-scale propagation of genetically similar plantlets, which further leads to the extensive propagation of this species. Tissue culture plays a crucial role in the preservation, propagation, and genetic improvement of *Eucalyptus* species. Despite unquestionable progression in biotechnological and tissue culture approaches, the productivity of plantations is still limited, often due to the low efficiency of clonal propagation from cuttings. The obtained F1 hybrids yield high biomass and high-quality low-cost raw material for large-scale production; however, the development of hybrid, clonal multiplication, proliferation, and post-developmental studies are still major concerns. This riveting review describes the problems concerning the in vitro and clonal propagation of *Eucalyptus* plantation and recent advances in biotechnological and tissue culture practices for massive and rapid micropropagation of *Eucalyptus*, and it highlights the *Eucalyptus* germplasm preservation techniques.

## 1. Introduction

*Eucalyptus* (Family: Myrtaceae) is a genus including rapidly growing evergreen ornamental trees and shrubs indigenous to Australia. Over 900 different species native to Australia, Argentina, Brazil, Chile, France, India, Indonesia, Portugal, Morocco, Spain, South Africa, and the USA have been planted, covering more than 20 million hectares of land [1,2]. The augmented demand for its pulp and hardwood as well as its adaptivity to various biotic stressors and high productivity have hiked its economic value [3]. Besides these, the essential oils concentrated in different organs of this plant possess biological, toxicological, and pharmacological applications, thus having a promising prospect for ethnomedicine [4]. Owing to the increased future demands, rapidly propagating *Eucalyptus* plantations are desired [5,6].

### 1.1. Challenges with Conventional Breeding

The conventional breeding programs involve the cultivation of *Eucalyptus* from seeds; however, challenges such as long generation time and genetic load persist [7]. These challenges have paved the way for modern biotechnological and plant tissue culture approaches, allowing for highly efficient rapid multiplication of plants in in vitro environments as well as germplasm storage and preservation [8]. Biotechnological tools, along with traditional tree improvement techniques, can be considered as solutions for meeting the growing human demand for forest products as compared to conventional tree improvement programs alone; also, biotechnology opens doors for understanding a wide range of complex biological problems related to forest tree species. Forest biotechnology covers a wide array of modern techniques covering aspects of genetic engineering, in vitro regeneration techniques, genomics, metabolomics, proteomics, molecular markers, and marker-assisted breeding. These advanced biotechnological tools have opened new doors for understanding the genetic structure, identifying biotic and abiotic interactions and stress tolerance factors, and linking some of the important genetic traits, thus facilitating accelerated selection and further breeding programs. One such efficient technique is micropropagation. The promising approach of micropropagation is achieved via organogenesis and somatic embryogenesis [9]. The history of *Eucalyptus* tissue culture can be traced back to the mid-20th century, and it has since become a significant tool in the conservation, propagation, and breeding of *Eucalyptus* species. Early experiments in the 1950s and 1960s faced challenges due to the limited knowledge of plant growth regulators and maintaining sterile conditions. However, the 1970s and 1980s marked a significant breakthrough, when pioneering researchers in Australia, such as Dr. Carron and Dr. Griffin, successfully developed protocols for the mass propagation of *Eucalyptus* species using tissue culture techniques. This success led to practical applications in the forestry industry, where tissue culture provided disease-free and genetically identical *Eucalyptus* plantlets for reforestation and commercial plantations. As techniques advanced, tissue culture became essential in *Eucalyptus* genetic improvement and breeding programs, focusing on enhancing desired traits such as growth rate and disease resistance. Cryopreservation techniques were also introduced to effectively conserve *Eucalyptus* germplasm. Despite challenges like somaclonal variation and disease outbreaks, ongoing research aims to improve tissue culture protocols for *Eucalyptus* species. *Eucalyptus* tissue culture continues to play a vital role in sustainable forestry practices, biodiversity conservation, and genetic preservation of valuable *Eucalyptus* species.

This paper attempts to discuss the methods of clonal propagation in upcoming sections. This study also focuses on the factors that influence the clonal propagation of *Eucalyptus* and further discusses the advances in germplasm preservation techniques. Previous research concerning traditional and modern breeding programs for improving *Eucalyptus* plantations in the last 25 years is included.

### 1.2. A Crop with Potential Biomass Production

As reported, nine dominating species of *Eucalyptus* have been cultivated for hardwood, namely *E. camaldulensis*, *E. pellita*, *E. dunnii*, *E. grandis*, *E. tereticornis*, *E. globulus*, *E. sargila*, *E. uropylla*, and *E. nitens* [10]. The F1 hybrids achieved from the breeding of these species show seed vigor and yield high biomass. Moreover, the F1 hybrids from interspecific breeding show heterosis. Successful attempts to achieve commercial-scale plantations have been made for some species (*E. deglupta*, *E. grandis*, *E. camaldulensis*, and *E. pellita*) agile to tropical and subtropical regions with high rainfall via macro- and microcuttings [10,11,12]. For those species unable to rapidly propagate using these methods, tissue culture is a propitious approach [13]. Events of natural hybridization between *Eucalyptus* species have been reported [14]. The spontaneous hybrid formation between *E. benthamii* and *E. dunnii* at Embrapa forests in Colombo, Paraná State, presented frost-tolerant varieties. Likewise, artificially pollinated hybrids serve as a viable alternative for future forest plantations while combining the useful traits from different parental combinations [15,16]. Initially, thirteen mature F1 *Eucalyptus* hybrid pairings were reported in India. Studies were initiated to create controlled and natural hybrids using half-sibling progeny generated from the seeds gathered from the stands of two intercross able species growing close to one another based on cross-ability patterns. The hybrids were generated mechanically or naturally through selection in the Dehradun campus of the Forest Research Institute’s New Forest area. Researchers had worked on two important interspecific F1 hybrids, namely FRI-5 (*E. camaldulensis* Dehn × *E. tereticornis* Sm) and FRI-14 (*E. torelliana* F.V. *Muell* × *E. citriodora* Hook), among several interspecific hybrids that were created [17]. The superiority of *Eucalyptus* hybrids FRI-5 and FRI-14 to their parentage has been established. Additionally, high levels of hybrid vigor have been observed in these plants, and they outperformed the parent combinations by three to five times in terms of growth characteristics. The average yield per unit area per unit of time was lower in the F2 population produced from seeds because of the significant segregation observed [17].

According to [18,19], these hybrids produced more biomass than their parents’ offspring and the Mysore Gum. The authors of these studies reported a micropropagation strategy for cloning and large-scale production as novel clonal propagation techniques for the commercial cultivation of these priceless *Eucalyptus* hybrids. The hybrid was observed to be intermediate to its parental species in more than half of the contrasting characteristics tested; this is noteworthy because of the two parent species involved, i.e., *E. grandis* and *E. tereticornis*. The former has a faster growth rate and good stem form, provides the best pulp quality, and prefers areas with high rainfall, while the latter is drought-tolerant, showing the suitability of hybrids toward intermediary environments (hybrid habitat). The outcomes of this hybrid micropropagation were reported for the first time in India. The species and hybrids that are reproduced via traditional methods (such as the many millions of *E. grandis* × *E. urophylla* and *E. grandis* plants produced per year) are often adapted to heavy rainfall areas in the tropics or subtropics. On drier, colder sites where land is more easily accessible and less expensive, hardwood plantations are being planted more frequently. Other *Eucalyptus* species are needed for these locations, such as species that are harder to propagate from cuttings, including *C. citriodora*, *E. globulus*, *E. cloeziana* F. Muell., or *E. nitens* [20,21]. The development of effective techniques for the clonal propagation of *Eucalyptus*, particularly for species that are difficult to propagate from cuttings, has remained a major issue in hardwood forestry. As tissue culture is among the most promising methods to rapidly propagate desirable genotypes and maintain germplasm in vitro, it has been used sustainably for these species and many others by various researchers [22].

## 2. Micropropagation and Its Applications

Achieving true-to-type plants with desirable traits using in vitro protocols is termed micropropagation (Figure 1 and Figure 2). Five critical stages need to be accomplished to successfully establish micropropagation [23]. These five stages include (i) Stage 0, defined as the preparatory stage for developing efficient and reproducible protocol; (ii) Stage 1, which aims to establish an aseptic and viable culture; (iii) Stage 2, which involves propagation without immolating objective to be achieved; (iv) Stage 3, involving large-scale propagation; and (v) Stage 4, the hardening stage of the established plant material. Each of the stages involved is discussed comprehensively in subsequent sections. Figure 3 illustrates different events of *Eucalyptus* micropropagation.

### 2.1. Establishing Axenic Culture

For ensuring the success of micropropagation, maintaining uncontaminated cultures are necessary throughout the protocol. The foremost important step in the in vitro propagation of plants is to establish a microbial contamination-free culture because the primary explants employed are nonsterile and thus a major source of microbial contamination in the culture [24]. The nodal segments bearing axillary buds, shoot tips, lead discs, and seeds could be used as explants to initiate tissue culture. The use of seeds as explants does not lead to true-to-type propagation; however, due to the ease of decontamination and juvenility of young seedlings, this method is superior for rapid clonal propagation [25,26]. Moreover, dormant axillary buds from nodes are widely used for maintaining clonal fidelity, providing thousands of plantlets via rapid propagation with high multiplication rates. The propagation from nodal segments is also reported in the micropropagation of woody plants like bamboo [27]. Nonetheless, the explants from leaf discs and shoot tips provide true-to-type propagation, although they are least preowned due to onerous sterilization [25,27]. The initial step involves the surface sterilization of the explant, which is usually achieved by first rinsing/washing the explant with nonsterilized water, immersing it in ethyl alcohol (70%), or treating it with chlorine-based sterilant such as calcium hypochlorite (Ca(OCl)_2_, mercuric chloride (HgCl_2_), and sodium hypochlorite (NaOCl), followed by rinsing with sterile distilled water. To prevent drying and increase the interaction between the explant surface and sterilant, a drop of wetting agents such as Tween 20 is often added. Although chlorine-based sterilants can efficiently sterilize the explants, their use is associated with toxic effects on living cells. Mercuric chloride has been reported to cause mammalian toxicity. So, the use of either calcium hypochlorite or sodium hypochlorite is recommended [28,29]. The basic explant sterilization of *Eucalyptus* is achieved by treating it with sodium hypochlorite at 67–1340 mM concentrations. However, the concentrations of sodium hypochlorite must be optimized before experimentation as the concentration affects the overall success of micropropagation. For instance, as reported, the seed germination of *Corymbia citridora* and *Corymbia torelliana* hybrid was decreased when the concentration of NaOCl was increased from 134 to 670 mM [30]. The optimization of the sterilization protocol of five different clones of eucalypt was performed using 1% sodium hypochlorite, 0.1% mercuric chloride, 1 mg/mL of rifampicin, and 70% ethanol as sterilant; the study suggested that, along with 1% sodium hypochlorite and 70% ethanol, using 0.1% mercuric chloride for 3 min was optimum, and using 1 mg/mL of rifampicin for 5 min was optimum for effective sterilization [31].

Leaching out of phenolics from cultured explants is also a main concern that prevents the establishment of in vitro cultures to a large extent. Reports suggest that nodal segments cultured on MS medium exudated phenolics. The number of phenolics exudated by nodal segments and bud-break response varied with the month of the collection of explants. Nodal segments were reported to be collected every month from January to December. It was found that explants collected during April and July to September showed the least phenolic exudation and better bud-break response comparatively and were best for in vitro studies. The phenolic exudation was high from October to January and May to June with poor bud-break responses [32].

### 2.2. In Vitro Proliferation of Shoot

In plant tissue culture, two main pathways are involved in achieving shoot regeneration: embryogenesis and organogenesis. Both processes play crucial roles in the development of new shoots, but they differ in the cellular mechanisms and morphological changes they undergo. The previous literature has only focused on “shoot multiplication”; however, for a better understanding, we provide a concise explanation for organogenesis and embryogenesis in the next sections.

#### 2.2.1. Organogenesis

Organogenesis refers to the process through which shoot regeneration occurs via the differentiation and development of new shoots from pre-existing meristematic cells or tissues. In this pathway, the initial step involves the induction of adventitious shoots from explants such as leaf, stem, or root segments. The explants are cultured on a nutrient medium supplemented with specific plant growth regulators, particularly cytokinin, which stimulate the formation of shoot primordia. These primordia then undergo further growth and differentiation to develop into shoots with organized structures, including leaves and stems. Organogenesis is often characterized by the initiation of multiple shoots, referred to as shoot proliferation or multiplication [33]. The formation of new organs directly from explants is termed direct organogenesis, while the formation of new organs from cell cultures or suspensions, tissues, or calluses is termed indirect organogenesis [33,34,35]. Furthermore, organogenesis also includes the regeneration of roots from explants, which is known as direct root organogenesis, wherein new roots are directly developed from explants. The development of new organs depends on a variety of factors, such as hormones and type of media [36]. The differential media and nutrients employed in *Eucalyptus* micropropagation are discussed in the next section. Mostly basal MS salts and Murashige Skoog media of different strengths are used for establishing organogenesis in *Eucalyptus*. Besides these, Woody Plant Medium; Driver and Kuniyaki Walnut medium; Juan, Antonio, Diva, and Salvin medium; Schenk and Hildebrandt medium; and B5 medium have also been extensively used [17,33,37].

#### 2.2.2. Somatic Embryogenesis

The development of bipolar embryos from somatic cells/tissues asexually is termed somatic embryogenesis [38]. It is a pathway for shoot, root, and plantlet regeneration that mimics the process of embryo development in plants that involves the dedifferentiation of somatic cells to a totipotent state, where they regain the ability to form an embryonic structure. Somatic embryos can be formed via two pathways, direct embryogenesis in which the embryo develops from pre-embryonic cells, or indirect embryogenesis in which the embryo develops from a callus grown on culture media. Pre-embryonic cells are undifferentiated cells that have the potency to differentiate into any kind of cells. Generally, apical meristems, hypocotyls, and epicotyls serve as a source of pre-embryonic cells as these contain undifferentiated cells [33,38]. A protocol for somatic embryogenesis has been reported by [39], suggesting that the leaves from adult trees are better for inducing somatic embryogenesis than floral tissues. The dedifferentiated cells progress through various stages of embryo development, including the formation of a proembryo, globular embryo, heart-shaped embryo, and mature embryo. Shoots arise from these embryogenic structures and can be further multiplied through subsequent subculturing. However, the inability of somatic embryos to reach the maturation stage is an adverse limitation of clonal propagation, the success of which is directly dependent upon the optimization of PGRs used, the age of tissue used, and the type of media used for establishment (Table 1). Typically, semi-solid MS media supplemented with sucrose are used to initiate *Eucalyptus* somatic embryogenesis; however, B5 media supplemented with sucrose have also been reported to induce somatic embryogenesis in *C. citridora*. Moreover, various PGRs have also been reported to induce embryogenesis in *Eucalyptus*; for instance, a hormone-free medium for inducing somatic embryogenesis was also suggested [40,41,42,43].

### 2.3. Adventitious Root Formation and Root Hardening

The bipolar structure formed after somatic embryogenesis can directly germinate by using nutrients from basal media for the shoot and root proliferation. Contrastingly, the unipolar structure needs the proliferation of adventitious roots at the base of their shoots for the development of plantlets. This is usually achieved via semi-solid media; often, activated charcoal is also introduced in these media as it regulates the pH of the media and is also reported to adsorb the inhibitory compounds from the media, in addition to reducing irradiance at the base of the shoots [57,58,59]. Improved in vitro rooting in *E. grandis* × *E. nitens* was reported by reducing the strength of MS media from full to half-strength and decreasing the concentration of sucrose in shooting media from 20 to 15 g L^−1^. Additionally, it was observed that increasing the concentration of NAA from 0.1 mg L^−1^ to 0.5 mg L^−1^ increased the average percentage of adventitious roots [29]. Also, increasing the IBA concentration from 0.1 mg L^−1^ to 0.5 mg L^−1^ increased root hair formation. Similar studies on *E. erythronema* × *E. stricklandii* suggested 8 weeks of continuous exposure to IBA on roots resulted in the longest root length [44]. Light studies on *E. grandis* × *E. urophylla* suggested the use of red–blue light to be superior for rooting and showed the highest mean number of roots [56,60].

Before adventitious root formation, the acclimatization of shoots is necessary for their future success in nursery conditions. Micropropagated plantlets were hardened using a liquid MS medium (1/4 strength) with 2% sucrose. Furthermore, for supporting roots, adsorbent cotton was soaked in this liquid medium. After maintaining for 2 weeks, the plantlets were transferred to mist bags containing a 1:1:1 ratio of soil, manure, and sand and then transferred to a net house. Finally, the plantlets were transferred to field conditions and showed 85–95% success rates in field conditions [17]. However, in another study, 58% of the success rate of plantlets in field conditions was due to the loss of some plantlets during handling [29]. Another innovative approach for acclimatization was reported, where the shoots were maintained in photoautotrophic culture at a high concentration of carbon dioxide and a low sugar concentration. These conditions promote carbon fixation and transpiration. Notably, 86–96% of success rate for *E. camaldulensis* and 100% success rate for *E. grandis* × *E. urophylla* have been achieved with this method [45,61,62,63]. A similar study suggested that increasing the temperature from 18–13 °C to 33–28 °C increased the number of root cuttings per stock plant [46]. Moreover, improved rooting efficiency was achieved in clones of *E. urophylla* via in vitro rejuvenation/reinvigoration [64].

## 3. Factors Affecting the Efficiency of Micropropagation

The efficiency of micropropagation can be influenced by various factors. Growth regulators play a crucial role in promoting shoot proliferation and rooting in tissue culture. The type and concentration of cytokinins and auxins in the culture medium can significantly impact the rate and quality of shoot formation. Additionally, the composition of the culture medium, including the types and concentrations of nutrients, vitamins, and carbon sources, affects the growth and development of explants. Organic elements such as amino acids and complex organic compounds can enhance shoot regeneration and plantlet growth, while inorganic elements, including macronutrients and micronutrients, are vital for overall plant health and development. Moreover, light is an essential environmental factor that affects micropropagation efficiency. Proper light intensity and photoperiod can influence shoot initiation, elongation, and rooting. The spectrum of light, particularly the red-to-far-red light ratio, also plays a role in regulating plant growth and differentiation. Optimizing these factors in micropropagation protocols is crucial for maximizing the efficiency of shoot proliferation and the production of healthy plantlets. Here, we discuss all these factors and their effects on *Eucalyptus* micropropagation.

### 3.1. Role of Plant Growth Regulators

Plant growth regulators or plant growth hormones are chemical compounds widely recognized to alter the growth of plants, for instance, to suppress the growth of shoots, boost the growth of shoots, or alter the maturity of the fruit. Indole-3-acetic acid (IAA) is a natural auxin in plants, and indole-3-butyric acid (IBA) is the analog of the auxin found in plants. Both IAA and IBA are synthesized via tryptophan-dependent or tryptophan-independent pathways [65,66]. Both IAA and IBA can be quickly metabolized in tissues of *Eucalyptus* plants. Since auxins play a crucial role in the regulation of cell division, as well as the elongation of plants and many other phases of their development, these are stored by plants as either IAA or IBA, which is converted into IAA whenever required [67,68].

Plant growth regulators have been reported to, directly and indirectly, influence plantlet proliferation, including the differentiation of embryos and different organs. For instance, optimal concentrations of cytokinin–auxin necessary for inducing organogenesis in *E. cloeziana* micropropagation were suggested by [47]. Moreover, the highest rate of somatic embryogenesis was also reported by introducing 0.1 mg L^−1^ NAA and 0.5 mg L^−1^ BA [48]. Increased shoot multiplication via the addition of 0.5 mg L^−1^ BAP in WPM and ½ MS medium was also reported [49]. It is important to note that the optimization of PGRs in culture media is necessary as they could negatively affect the growth of plantlets [49,69]. Similar studies on auxin types in *E. salgina* and *E. globulus* were reported, suggesting the best rooting obtained with IBA than IAA. Best rooting was achieved in both species when treated with IBA; the possible explanation supporting this IAA is that it is highly susceptible to enzymatic degradation and is also 5 times more susceptible to photo-oxidation than IBA. In support of this, a similar study on *E. sideroxylon* micorcuttings was reported by comparing IBA and NAA using different concentrations of both auxins from 0 to 10 μM. A high frequency of callus induction was reported by culturing cotyledon explants on MS media supplemented with 1 mg L^−1^ NAA + 0.5 mg L^−1^ BA. The same study also suggested that MS media supplemented with 0.5 mg L^−1^ NAA + 1 mg L^−1^ BA + 1 mg L^−1^ GA_3_, as well as ½ MS media supplemented with 0.5–1 mg L^−1^, resulted in high-frequency adventitious root formation in *E. bosistoana.* Furthermore, 100% survival of preacclimatized plantlets was obtained [70]. A similar study revealed best shoot elongation by supplementing the media with 0.05 mg L^−1^ BAP + 1 mg L^−1^ NAA and 0.05 mg L^−1^ BAP + 1 mg L^−1^ NAA + 1 mg L^−1^ IBA^−1^ [71]. The results showed increased callus induction in micorcuttings exposed to IBA compared with the micorcuttings exposed to NAA. However, the responses to auxins may vary from species to species based on their differential affinities, uptake, and metabolization of auxins [72,73].

### 3.2. Effect of Culture Media

The initiation of shoot proliferation is usually achieved by culturing explant on a semi-solid medium that comprises gelling agents such as 1.5–4.0 g/L of gelrite, 4–8 g/L of agar, or 1.5–4.5 g/L of phytagel, and the pH is adjusted between 5 and 6. Also, the use of liquid media has been reported for the initiation and proliferation of nodes and shoots [74,75]. The culture of the shoot depends on the ability to encourage the development of axillary and accessory buds that are present at the base of each leaf axil. In previous research attempts at *Eucalyptus* micropropagation, the basal media used include Murashige and Skoog media (different strengths ½, ¼); JADS (Juan, Antonio, Diva, and Silvian) medium, WPM (Woody Plant Medium), and DKW (Driver and Kuniyaki Walnut) medium [76]. Additionally, the form of medium, whether a liquid suspension medium or a semi-solid medium, affects the growth of plantlets in vitro. Reportedly, better shoot multiplication of *Eucalyptus* has been observed in liquid media than in semi-solid media [77]. WPM was reported as the optimal medium for the micropropagation of *E. benthamii* [50]. Similarly, the JADS medium was observed to be optimal for the trunk base shoot elongation of *E. grandis.* The DKW medium was used as an alternative for the micropropagation of *E. nitens* [51]. The adjustment of the pH of the medium to 5.8 prior to autoclaving at 121 °C for 15 min has been recommended. Temperature incubation at 25 ± 2 °C and 16 h photoperiod with the photon flux density of 2500 lux from white fluorescent tubes is the recommended cultured condition for *Eucalyptus* spp. For improving the survival of the explant, polyvinyl pyrrolidone, activated charcoal, and ascorbate have also been added to culture media. However, the type of media and the response of plantlets vary among species of *Eucalyptus* and their hybrids. Moreover, some drawbacks such as chlorosis, tissue browning, and oxidation have been reported in almost all types of media used for *Eucalyptus* micropropagation.

### 3.3. Importance of Organic and Inorganic Elements

Organic and inorganic elements play a crucial role in plant micropropagation media, which are used for the propagation and growth of plants under sterile conditions. Elements like calcium (Ca), nitrogen (N), phosphorous (P), and boron (B) serve as macro- and micronutrients essential for nourishing plant growth. These have been introduced in in vitro cultures to promote the proliferation and differentiation of organs from the shoot. An appropriate balance and concentration of these organic and inorganic elements are crucial to ensure the successful propagation and growth of plants in vitro [78]. The composition of the medium can be adjusted based on the specific requirements of different plant species and their growth stages. The elements required by plants in concentrations lower than 0.5 mM/L are referred to as micronutrients, and the elements with more than this concentration are referred to as macronutrients [79,80,81]. Magnesium (Mg), calcium (Ca), hydrogen (H), sulfur (S), potassium (K), nitrogen (N), phosphorous (P), and oxygen (O) serve as macronutrients. Calcium act as an important cofactor and cellular messenger involved in various signal transduction pathways and is well known to play various important roles in plant stress [82]. For instance, in *E. urophylla* and *E. grandis*, calcium has been reported to trigger organogenesis [83]. Manganese (Mn), chlorine (Cl), iron (Fe), zinc (Zn), boron (B), sodium (Na), iodine (I), and copper (Cu) serve as the microelements among which iron is the most critical element. Also, it was reported that the deficiency of boron in media led to necrosis and callus accumulation, further inhibiting seedling growth in *E. grandis* [84]. Furthermore, a study suggested that calcium, in the form of calcium chloride in agitated liquid media, decreases the hyperhydricity in *E. saligna*. However, due to the toxicity caused by chlorine, it had been not effective in completely eliminating hyperhydricity [85]. Calcium chloride dihydrate was also reported to induce shoot elongation and decrease vitrification [86]. Nonetheless, efforts in improving the optimization and choice of organic/inorganic elements have been increased, although they are insufficient in considering the individual role of vast available macro- and micronutrients.

### 3.4. The Role of Carbohydrates

Carbohydrates are important biomolecules that provide biofuel and serve as a carbon source for cell growth. Different reducing and nonreducing sugars are available and have been employed in micropropagation like glucose, fructose, sucrose, and galactose, among which sucrose is still the most preferable in *Eucalyptus* micropropagation due to its ease of translocation in plant tissues. Sucrose is a nonreducing sugar, specifically a disaccharide composed of fructose and glucose. Some of the reports suggest that increased sucrose concentrations can hinder water and nutrient uptake in plants and inhibit photosynthesis by influencing photosynthetic enzymes. However, contrastingly, some studies suggest that plants remain uninfluenced by high sucrose concentrations. Furthermore, in *E. cloiziana*, high glucose and sucrose concentrations were reported to decrease the shoot length (conc. > 15 g/mL in media) [87]. For instance, studies reported different concentrations of sucrose (1–6%) in MS media for in vitro shoot proliferation. The best results were reported using 3% sucrose in MS media with a 6–7-fold increased shoot multiplication. Similar findings reported that media devoid of sucrose result in the inhibition of shoot multiplication, and the leaves and shoots turned to a pale green color [88,89]. The results of this study are in line with those of several studies that used 3% sucrose as a carbohydrate source to promote the growth of shoots in a variety of *Eucalyptus* species. However, similar findings on many other woody plants have also been reported; for instance, in bamboo shoots, successful multiplication was observed when media were supplemented with 2% sucrose [90,91]. It has also been reported that an increase in sucrose levels from 3 to 4% does not cause any effect on shoots but results in albinism. Similarly, at 1% sucrose concentration, thin shoots and leaves were developed that were inappropriate for subculturing. An investigation was conducted on myo-inositol to determine its role in in vitro shoot multiplication. MS media supplemented with 100 mg L^−1^ of myo-inositol yielded the best shoot multiplication rates, while MS media devoid of myo-inositol showed decreased shoot multiplication. Moreover, MS media supplemented with excess myo-inositol (more than 150 mg L^−1^) not only decreased shoot multiplication but also had detrimental effects on shoots. For maximizing shoot multiplication rate and growth, 100 mg L^−1^ myo-inositol was supplemented in culture media for all trials [6,88].

### 3.5. Effects of Radiation and Light Exposure

Light is a critical external aspect influencing the different phases of plant growth. Light hour durations and intensity are directly linked to plants’ photosynthetic rates [92]. Many hybrids have been studied that suggest the effects of light and radiation on the success of micropropagation. The effects of five sources of lights, namely fluorescent lamps, white LEDs, red LEDs, blue LEDs, and red–blue LEDs, on *E. grandis* × *E. urophylla* hybrid were studied, and red–blue LEDs and florescent lights were found to be superior for *E. grandis* × *E. urophylla* micropropagation. The response of *Eucalyptus* to micropropagation varies among genotypes. Moreover, a low level of irradiation triggered rooting in *E. globulus*; contrastingly, some studies confirmed that low-level irradiations hindered root proliferation in *E. globulus* [52,53,60]. Also, studies on *E. salgina* and *E. globulus* were conducted for the effect of light on rooting capacity using white fluorescent lamps. *E. globulus* did not show any effect from exposure and was found to be dependent only on exogenous auxin concentration for rooting, while *E. salgina* cuttings showed increased root density per rooted cutting upon exposure to light combined with exogenous auxin application [93]. A similar study suggested that preservation under low light intensity effectively preserved cultures for 3 months [54]. Besides these, increases in light intensity and carbon dioxide content have been shown to increase the growth of explants photoautotrophically [94]. In a similar study, the effect of light quality on the clone of *E. urophylla* in the photoautotrophic system was assessed, focusing on the stomatal density, carotenoid content, chlorophyll content, the number of shoots, and the longest shoot. The results indicated that blue LED resulted in fewer shoots, while high production of carotenoids was observed under white light [95]. In another study on *Eucalyptus dunnii* and *Eucalyptus grandis* × *E*. *urophylla*, it was observed that the use of white light was associated with increased buds per plant, decreased tissue oxidation, and longer shoot length. In *E. dunnii*, blue, red, and yellow light resulted in increased chlorophyll a and b content. Also, blue, white, purple, and red light increased stomatal densities. Moreover, a previous study revealed that irrespective of light spectra, *E. dunnii* showed decreased adventitious rooting [96]. Another similar study on *E. grandis × E. urophylla* clone was carried out to assess the impact of five different light sources, namely fluorescent lamps as well as blue, green, red, and yellow cellophane light in a bioreactor system. Yellow and blue light sources were found to be more suitable for the clone as less hyperhydricity was observed along with spongy parenchymatic tissue, thicker mesophyll, increased shoot length, and more shoots per explant [97]. Moreover, another study suggests that for the in vitro multiplication of *E. pilularis*, white light was more suitable, and for the *E. urograndis* clone, blue light was more suitable because it increased the number of buds, shoots length, and fresh weight per explant [98]. Table 1 and Table 2 highlights critical factors in *Eucalyptus* micropropagation.

## 4. In Vitro Germplasm Preservation

In vitro germplasm preservation plays a vital role in conserving plant biodiversity, protecting endangered species, and safeguarding important genetic resources for future research, breeding programs, and restoration efforts. It helps to maintain a diverse and resilient gene pool, ensuring the availability of plant materials for sustainable agriculture, forestry, and environmental conservation. Brief insights into recent preservation techniques that are cryopreservation and cold storage in *Eucalyptus* spp. are discussed below.

The preservation of germplasm is peremptory for breeding programs. Cryopreservation is a protocol that involves germplasm storage at ultra-lower temperatures (−135 °C to −196 °C) in liquid nitrogen. At this temperature, the cell viability and genetic stability are preserved; however, the aging of the cell is hindered due to the halting of all biochemical and physiological pathways of the cell [99]. Various reports on successful cryopreservation of *Eucalyptus* spp. have been reported, including shoot tips of *E. grandis* × *E. camaldulensis*, *E. urophylla* × *E. grandis*, *E. grandis*, *E. grandis* × *E. urophylla*, and *E. camaldulensis* using the droplet vitrification method, firstly by preculturing shoot tips on MS media containing 0.25 M sucrose concentration for 24 h and then again on MS media containing 0.625 M sucrose concentration for 24 h, followed by subjecting the individual shoot tips to plant vitrification solution (PVS). Similarly, the preservation of the axillary buds of *E. grandis* × *E. camaldulensis* has also been reported by placing the axillary buds in semi-solid MS media and significantly increasing the concentration of sucrose and glycerol from 0.4 to 0.7 to 1.0 M; this type of method showed 49% of regrowth [55,99,100].

Cold storage involves the preservation of plants at lower temperatures. Some attempts have been made to preserve shoots of *E. grandis* in cold storage using half- and full-strength media. However, half-strength MS media allowed for the preservation of shoots at 24 °C to 28 °C for 10 months; contrastingly full-strength media could preserve the same shoots at 10 °C only for 6 months. Similar studies were conducted on *Corymbia toleriana* and *Corymbia citridora* and showed the preservation of shoots on full-strength MS media at 14 °C for 12 months [101].

## 5. Limitations, Challenges, and Future Directions

Even though successful attempts in micropropagation have been made, some challenges like hyperhydricity, phenolic oxidation, explant contamination, and root and ex vitro survival of the established clone still persist. Addressing these problems requires more effort and refinement of tissue culture protocols, including the optimization of growth media, hormone formulations, and culture conditions. Strict aseptic techniques and effective sterilization methods are crucial to minimizing endogenous contamination. Moreover, future directions in exploring novel approaches such as genetic engineering for improving the performance of propagated clones and utilizing “muti-omics” approaches for understanding the molecular basis of various stress, responses, and key regulators affecting the overall efficiency of propagated plants can help in enhancing the success of micropropagation. This comprehensive review holistically addressed the gaps in the large-scale propagation of the *Eucalyptus* plants, thus helping the scientific community with further research in a useful direction. Overcoming these challenges will contribute to the successful large-scale propagation of forest species through in vitro techniques, which will benefit reforestation efforts and conservation initiatives.

## 6. Conclusions

Micropropagation has allowed for achieving rapid clonal plantations; however, the efficiency and success of micropropagation depends on the rate of shoot multiplication. *Eucalyptus* spp. and its hybrids are among the commercially important tree spps and have been extensively propagated via micropropagation. These are continuously studied for their genetic improvement and establishment of superior hybrid species. The micropropagation of *Eucalyptus* can be achieved in four main steps: (i) suitable explant collection, (ii) the preparation of aseptic culture, (iii) shoot and root proliferation, and (iv) the hardening of the root. A wide range of media, both semi-solid and liquid culture media, have been described for the efficient clonal propagation of *Eucalyptus*. Currently, many studies are still being conducted for the development of improved tissue culture protocols as well as for improving the wood quality of *Eucalyptus*. For achieving success in the callogenesis and histogenesis of *Eucalyptus* spp., complete knowledge of all optimization procedures, including the type of media, as well as the choice of hormones, their concentrations, and ratios, must be known. Apart from that, the age and type of tissues, as well as the seasons during which explants are collected, are necessary to consider for ensuring the success of tissue proliferation. Moreover, the field success of plantlets is necessary, for which proper conditions and protocols for the acclimatization of plantlets must be known. All of these important factors have been compiled in the present study. However, more studies are required to close the gap between the in vitro development of plantlets of *Eucalyptus* and their growth in field conditions, which is important for its industrial success.

## Figures and Tables

**Figure 1 plants-12-03018-f001:**
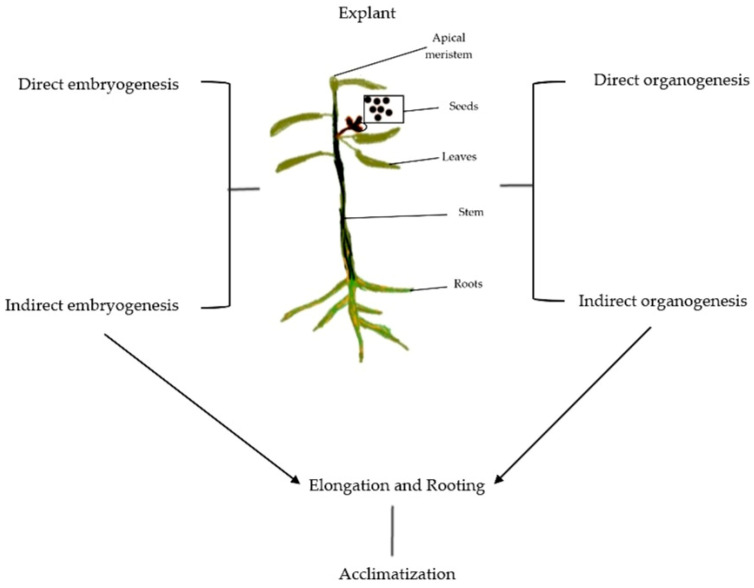
Schematic illustration of *Eucalyptus* micropropagation presenting sources of explant and basic steps involved in the protocol.

**Figure 2 plants-12-03018-f002:**
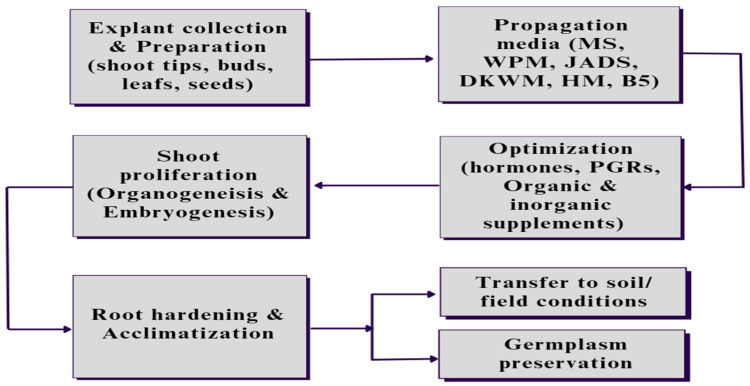
The steps involved in *Eucalyptus* micropropagation start with the collection and preparation of explants, which could be an axial leaf, shoot tips, buds, or seeds, followed by inoculating and culturing in specific media with the optimized concentration of the hormones and other nutritional supplements based on the requirement of the plant as well as environmental conditions. After shoot proliferation, the developed plants are subjected to root proliferation and root hardening in field conditions, or the germplasm of the developed plants could be preserved for future use in similar related protocols.

**Figure 3 plants-12-03018-f003:**
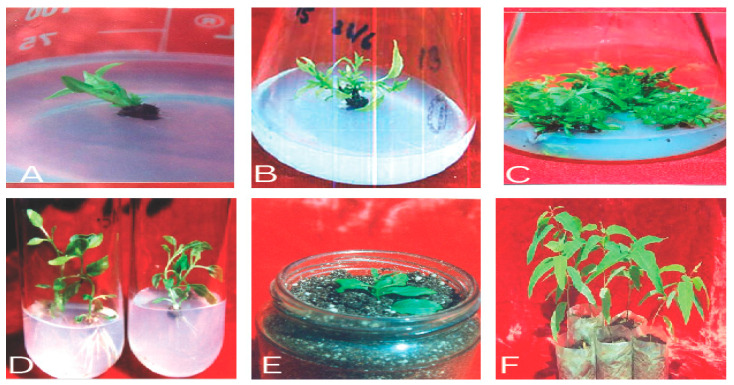
(**A**–**F**) Different stages in *Eucalyptus* micropropagation: (**A**) aseptic shoot/explant inoculation; (**B**) aseptic culture establishment; (**C**) shoot proliferation; (**D**) in vitro rhizogenesis of microshoots; (**E**) hardening; (**F**) acclimatization and field transfer.

**Table 1 plants-12-03018-t001:** The table below highlights various sources of explants, culture media, additives, etc., that have been previously used in *Eucalyptus* micropropagation.

Explant-Source/Type	Culture Media/PGRs/Additives	Experimental Outcomes/Remarks/Productivity/Root Hardening, etc.	References
Nodal segments	MS media, WPM, 0.5 mg L^−1^ IBA, 1.0 mg L^−1^ BAP, B5, NAA, Kinetin	Full-strength MS media supplemented with 1.0 mg L^−1^ BAP showed the best shoot elongation; ½ MS media supplemented with 0.5 mg L^−1^ IBA showed the best rooting; the resulting hybrid produced yielded 3–5 times more wood than parental species.	[5]
Nodal segments (30–32-year trees)	MS media, WPM, B5 medium, SH medium, IBA, BAP, NAA,	Best in vitro studies were obtained from the explants collected during the period of January to February and August to September; ½ MS media + BAP resulted in best rooting (92%); up to 98% of plants survived acclimatization.	[6]
Stem cuttings and axillary buds	MS Media, 2.22 µm BA, 1.16 µm Kinetin, 0.029 µm gibberellic acid, 400 mg L^−1^ PVP, 30 g L^−1^ sucrose	Micropropagation and microcuttings showed higher adventitious rooting (24.8–100% and 43–95%, respectively) than stem cuttings (9.3–75.5%).	[12]
Maintained Elite clones (KE8, CE2, T1, and Y8)	Basal MS media, 2.5 µM BA, 0.5 µM NAA, D-mannitol (0, 250, 500, 750, and 1000 mM)	The culture growth index of all clones was reduced significantly because of drought stress.	[17]
Cuttings from plants grown in vitro	Full and ½ strength MS media	Concentrations of IAA, IBA, and stem anatomy had no effect on the rooting potential of shoots.	[22]
Nodal cuttings	IAA, IBA (0, 1, 3, 8 g Kg^−1^)	The position from which the explant is harvested can affect the rooting potential and seed vigor. Explants obtained from 7/8 and 9/10 apical positions showed enhanced rooting and shooting.	[25]
Cuttings from 6-month-old parental plants	MS media, IAA, meta-topolin, kinetin, BAP, vit. B5, biotin, sucrose	0.5 mg L^−1^ meta-topolin and 1 mg L^−1^ IAA enhanced shoot elongation as well as bud proliferation, while 0.5 mg L^−1^ IAA resulted in the most consistent rooting percentages. Moreover, equal expression of *AUX1* and *PIN1* transporter genes increased responsiveness toward PGRs.	[29]
Nodal segments	MS media, 58 mM sucrose, 0.5 μM NAA, 2.5 μM BA,	Media supplemented with 1.0 μM 2,4-D, 5.0 μM BA and 500 mg L^−1^ cefotaxime showed maximum (44.6%) shoot bud organogenesis.	[33]
Nodal segments	MS media, 2 mg L^−1^ BAP, 0.1 mg L^−1^ NAA	Media supplemented with 0.5 mg L^−1^ showed the best shoot elongation, ½ MS + 1 mg L^−1^ IAA showed the best root induction and elongation, and direct regeneration was observed in MS + 20:1 BAP: NAA.	[34]
Young shoot segments	WPM, MS media, 2iP, NAA, BAP, sucrose	Media supplemented with BA resulted in 99% shoot proliferation, media supplemented with 2iP resulted in 93% shoot regeneration, and IBA promoted rooting in 60% of the clones.	[35]
Nodal segments	EDM basal media (a novel basal media for *E. dunnii*) supplemented with 20 g L^−1^ sucrose and without PGRs	Higher Fe, Cu, Zn, and Mn concentrations in EDMm media increased rooting. Moreover, high S and K concentrations in EDMm increased growth rate and multiplication. Also, no Fe chlorosis/oxidation was observed in shoots cultured on EDMm.	[37]
Zygotic embryo	One of the following media and growth regulators: ½ or full-strength MS media/WPM/B5/DKW/JADS media/3 mg L^−1^ NAA/10 mL^−1^ silver nitrate/0.5 mg L^−1^ DTT/100 mg L^−1^ ascorbic acid/0.5 mg L^−1^ DTE/1% m/v PVP/1% m/v PVPP/0.01% *w*/*v* activated charcoal	The best media for somatic embryogenesis were B5 and MS. Moreover, Silver nitrate, activated charcoal, and DTE reduced the browning of explants.	[41]
Somatic embryos	MS media supplemented with 3 mg L^−1^ NAA	MS medium without PGRs is highly efficient for promoting cotyledonary embryo proliferation and germination.	[42]
Zygotic embryo cotyledons	Hormone-free MS media	The reserve accumulation of mature zygotic embryos was analyzed. Cotyledonary somatic embryos possess a low density of starch and no lipids/proteins.	[43]
Axillary shoots	½ MS, 4.4 μM, 1 μM NAA, 1 g L^−1^ sucrose	WPM and QL media supplemented with Gibberellic acids showed enhanced shoot proliferation, ½ WPM supplemented with 20 μM IBA showed enhanced rooting, and 67% Plantlet hardening was achieved.	[44]
Shoot segments	MS media, 0.02 mg L^−1^ IBA	Vitron vessel placed in Low Photon Flux density at 3000 ppm CO_2_ for 24 h/day yielded the best growth and quality of plantlets.	[45]
Seedlings grown from seeds	Variable potting mixture	Low temperatures of 18 °C/13 °C to 23 °C/18 °C (day/night) reduced the number of harvested cuttings; however, they did not affect the percentage of roots proliferated from cuttings. By contrast, increasing the temperature to 33 °C/28 °C resulted in an increased number of cutting per stock plant.	[46]
Hypocotyl segments and cotyledonary leaves	MS media supplemented with different concentrations of NAA and TDZ, 0.8 g L^−1^ PVP, 0.1 g L^−1^ biotin, 0.1 g L^−1^ calcium pantothenate, 30 g L^−1^ sucrose	0.44 µM BAP increased the regeneration of adventitious buds.	[47]
Zygotic embryos and cotyledons	MS media supplemented with 3 g L^−1^ sucrose and different concentrations of NAA, 2,4-D, BA, ABA	1 mg L^−1^ NAA resulted in maximum callus induction, the frequency of callus proliferation depends on the age of the explant, with 10-year-old explants showing maximum proliferation, the highest frequency of somatic embryogenesis was observed in callus from mature zygotic embryos, low ABA concentrations increased number of somatic embryos.	[48]
Nodal segments	1/2 MS supplemented with different concentrations of BAP, NAA, and GA_3_	0.050 mg L^−1^ BAP achieved optimal bud proliferation + 0.50 mg L^−1^ NAA, while ½ MS media supplemented with 0.2^−1^ and 0.10 mg L^−1^ GA_3_ + 0.10 mg L^−1^ BAP showed highest shoot elongation.	[49]
Nodal segments	MS media without PGRs	Media free from GA_3_ + BAP resulted in best shoot elongation, and WPM + 0.05 mg L^−1^ NAA + 0.5 mg L^−1^ BAP resulted in maximum axillary bud proliferation.	[50]
Nodal segments	½ MS media, De-Fossard Medium, 0.9 µg L^−1^ BA, 0.5 µM NAA	The best multiplication rate (2.25) was achieved, and 93% of the plants survived acclimatization.	[51]
Nodal segments	MS media supplemented with 0.05 μM NAA, 0.4 μM BA, 1 mg L^−1^ nicotinic acid, 1 mg L^−1^ pyridoxine-HCl, 1 mg L^−1^ thiamine, 2 mg L^−1^ glycine, 50 mg L^−1^ myo-inositol, and 30 g L^−1^ sucrose	Endogenous rhythms cause time-related fluctuations, resulting in rooting variations among closely related genotypes.	[52]
Epicotyl segments	½ MS supplemented with 1/6× CaCl_2_, 2% (*w*/*v*) sucrose	Auxin reduced mean rooting time, and light conditions did not affect the rooting efficiency; with increased age, decreased rooting capability was observed.	[53]
Axillary buds	½ MS supplemented with 1 g L^−1^ ABA	Encapsulation by calcium alginate and storing under low light intensities resulted in the preservation of cultures for up to 3 months without affecting their viability.	[54]
Apical shoots	MS media supplemented with 0.04 mg L^−1^ BA, 1% sucrose, with/without charcoal	38–85% survival was observed with plants exposed to PSV2 for 30 min in liquid nitrogen.	[55]
Nodal segments	MS media supplemented with 30 g L^−1^ sucrose	The best in vitro establishment, multiplication, shooting, and rooting were achieved using red–blue LEDs and fluorescent lamps.	[56]

Abbreviations: 2,4-D—2,4-dichlorophenoxyacetic acid; ABA—abscisic acid; BA—6-benzyladenine; BAP—6-benzylamino purine; CaCl_2_—calcium chloride; CO_2_—carbon dioxide; Cu—copper; DTE/DTT—dithioerythritol; DKW—Driver and Kuniyaki Walnut; EDM—*E. dunnii* basal medium; Fe—iron; GA3—gibberellic acid; HCl—hydrochloric acid; IAA—indole-3-acetic acid; IBA-indole-3-butyric acid; JADS—Juan, Antonio, Diva, and Silvian; K—potassium; LED—light-emitting diode; Mn—manganese; NAA—α-naphthaleneacetic acid; PGRs—plant growth regulators; PSV2—plant vitrification solution 2; PVP—polyvinylpyrrolidone; PVPP—polyvinylpolypyrrolidone; QL—Quoirin and Lepoivre; S—sulfur; SH—Schenk and Hildebrandt; TDZ—thidiazuron; WPM—Woody Plant Medium; Zn—zinc.

**Table 2 plants-12-03018-t002:** The table below highlights the composition of the media, additives, and sterilants used in various successful attempts regarding *Eucalyptus* hybrid production.

Species	Explant	Sterilant	Media; PGR (If Any)	Area Studied and Scope of Work	References
*E. camaldulensis* × *E. tereticornis* and *E. torelliana* × *E. citriodora*	Nodal segments from mature trees (30–32 yrs)	0.15% HgCl_2_	MS media, WPM, SH medium, B5 medium; BAP, NAA	Hybridization of *Eucalyptus* species. The study reported that two hybrids developed that showed superior performance than parental genotypes.	[17]
*E. grandis* × *E. nitens*	Axial buds	10 g L^−1^ CaOCl	MS media; BAP, IAA, metatopolin, kinetin	Individual evaluation of each stage of micropropagation. The study reported that Auxins are principal components of media, and expression of different auxin transporters might be used as markers to identify *Eucalyptus* spp. amenable for micropropagation.	[29]
*E. erythronema* × *E. stricklandii*	Seedlings germinated in vitro	3% NaOCl	MS media supplemented with sucrose 30 g L^−1^; IBA, NAA, Gibberellic acids	First micropropagation report of ornamental *Eucalyptus* spp. The study reported that successful micropropagation from juvenile seedlings was achieved.	[44]
*E. benthamii* × *E. dunni*	Nodal segments from 1-year-old plants	NaOCl	½ strength MS media; PVP40, NAA, BAP	Optimization of chlorine concentration for explant sterilization and optimum ratio of PGRs for shoot elongation. The study reported that 0.5% NaOCl is suggested for nodal segments; 0.50 mg L^−1^ BAP + 0.05 mg L^−1^ NAA provides the highest number of bud proliferation.	[49]
*E. erythronema* × *E. stricklandii*	Nodal segments	1% NaOCl	MS media supplemented with sucrose 30 g L^−1^; 0.05 μM NAA and 2.22 μM BAP	Effect of different light intensities on micropropagation efficiency. The study reported that red–blue LEDs and fluorescent light result in higher vigor, high photosynthesis, and increased shoot and root proliferation.	[56]

## Data Availability

Data are contained within this article.

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
