# Peer review of "A Comprehensive Review Uncovering the Challenges and Advancements in the In Vitro Propagation of Eucalyptus Plantations"

_plants, 2023, doi:10.3390/plants12173018_

Round 1

Reviewer 1 Report

Article “A comprehensive review unleashing the challenges and advancements in in-vitro propagated Eucalypt plantations” is written well but some major information is still missing.

1.       What’s the importance of this article and it should be described in abstract and in the last Para of introduction and reasons for tissue culture

2.       Add History of eucalypt tissue culture

3.       2.2.1 Organogenesis: add different compounds and regulators

4.       The key role of plant growth regulators should be added

5.       Can we propagate eucalyptus  by seeds and cuttings

6.       Tree Improvement And genotype conservation should be discussed

7.       Limitation, challenges and future direction should be added in separate headings.

8.       There are 101 references out of which only 20 references are from 2020- 2023. Most of the references are too old that’s acceptable. At least 50% should be in between 2020- 2023.

9.       Ref no.  50 is incomplete.

10.   References are not uniform. There font and size varies greatly.

11.   Provide some illustration for propagation scheme.

minor improvement needed. 

Author Response

Thank you Sir/ Ma’am for your suggestions, considering your valuable feedback report, we have updated the manuscript as follows and highlighted in yellow.

Reviewer 1

Comments and Suggestions for Authors

Article “A comprehensive review unleashing the challenges and advancements in in-vitro propagated Eucalypt plantations” is written well but some major information is still missing.

  1. What’s the importance of this article and it should be described in abstract and in the last Para of introduction and reasons for tissue culture. Corrected
  2. Add History of eucalypt tissue culture. Corrected
  3. 2.2.1 Organogenesis: add different compounds and regulators. Added in the table 1.
  4. The key role of plant growth regulatorsshould be added. Corrected
  5. Can we propagate eucalyptus  by seeds and cuttings. Added in table 1, 2
  6. Tree Improvement And genotype conservation should be discussed. Corrected.
  7. Limitation, challenges and future direction should be added in separate headings. corrected
  8. There are 101 references out of which only 20 references are from 2020- 2023. Most of the references are too old that’s acceptable. At least 50% should be in between 2020- 2023. We have tried to add all the possible references that could fit in the requirements and directions of this review.
  9. Ref no.  50 is incomplete. Corrected
  10. References are not uniform. There font and size varies greatly. Corrected
  11. Provide some illustration for propagation scheme. Corrected

Reviewer 2 Report

Authors did a review on A comprehensive review unleashing the challenges and  advancements in in-vitro propagated Eucalypt plantations. Overall manuscript is well prepared. Some of the minor changes are
1. Abstract should be more discriptive.

2. Some more figures should be added.

3. Check for the uniform formating of headings and subheadings.

4. what is the novelty of this review.

Minor editing of English language required

Author Response

Thank you Sir/ Ma’am for your suggestions, considering your valuable feedback report, we have updated the manuscript as follows. We have highlighted the corrections in yellow.

Reviewer 2

Comments and Suggestions for Authors

Authors did a review on A comprehensive review unleashing the challenges and  advancements in in-vitro propagated Eucalypt plantations. Overall manuscript is well prepared. Some of the minor changes are
1. Abstract should be more descriptive. Corrected

  1. Some more figures should be added. Corrected (Figure 1, 3 added)
  2. Check for the uniform formating of headings and subheadings. Corrected
  3. what is the novelty of this review. Corrected (in last para of introduction)

Reviewer 3 Report

This manuscript reviewed the progress in in-vitro propagation of Eucalyptus spp. and its uses in Eucalyptus germplasm preservation. Generally, It is appropriately organized, but it encounters some problems as followings. 

Major concerns:

(1) In ‘2.2.1 Organogenesis’, the authors indicated organogenesis refers to the process of shoot regeneration. Actually, organogenesis should include the regeneration of roots, and this part should be described in the manuscript.

(2) In ‘2.2.2 Somatic embryogenesis’, the authors indicated it is a pathway for shoot regeneration. Somatic embryogenesis is also the pathway for root or plantlet regeneration.

(3) Line 253, the authors proposed that somatic embryos may directly develop from pre-embryonic cells. Considering the cultured explants are usually leaves, stems and roots, the sources of pre-embryonic cells should be described.

(4) Figure 2 provided little information, and the items are repeated, for example, ‘Carbohydrates’ is also ‘Organic and inorganic supplements’ and ‘Culture media’, while the above contents have been described in the text. This figure could be deleted.

(5) As to ‘4. in-vitro germplasm preservation’, there is only one sub-title (4.1 cryopreservation and cold storage) in this part, and this sub-title could be deleted. On the other hand, the authors should pay more attention to the writing format of all of the topic titles in the manuscript.

(6) Table 1 and Table 2 should be in the form of three-line table, and the table title should be put above the table. The abbreviations could be put under the table.

Minor concerns:

Line 25, High-quality: high-quality

Line 3, 28, 30, Eucalypt: Eucalyptus

Line 46, 68, 81, Eucalyptus: Eucalyptus (italicized)

Line 67, Traditional and Modern: traditional and modern (not capitalized word)

Line 86, 87, Tropical and Subtropical: tropical and subtropical (not capitalized word)

Line 90, and: and (not italicized)

Line 130, Table 2, Line 156, Line 169, Eucalypt: Eucalyptus

Line 155, Additives: additives (not capitalized word)

Line 161, Five: five

Line 164, 165, stage: Stage

Line 185, Dormant Axillary: dormant axillary

Line 193-207, it is not suitable to capitalize the name of chemical reagents.

Line 204, the full genus name for C. citridora should be provided.

Line 213, Nodal: nodal

Line 242, 244, 264, 267, Eucalypt: Eucalyptus

Line 273, Semisolid: semisolid

Line 284, Red: red

Line 320, Eucalypt: Eucalyptus

Line 329, Tissues of Eucalyptus: tissues of Eucalyptus plants

Line 341, Auxin: auxin

Line 346, Comparing: comparing

Line 362, Gelling: gelling

Line 367: Eucalypt: Eucalyptus

Line 379, eucalyptus: Eucalyptus

Line 382, Eucalypts: Eucalyptus

Line 384, Eucalypt: Eucalyptus

Line 390-419, 440, 442, it is not suitable to capitalize the name of elements and chemical reagents.

Line 420, Eucalypt: Eucalyptus

Line 433, to promote. The: to promote the

Line 434, Eucalyptus: Eucalyptus (italicized)

Line 454, 455, Use of Red/ Blue LEDs and Florescent: use of red/ blue LEDs and florescent

Line 456, Eucalypt: Eucalyptus

Line 460, White: white

Line 466, Carbondioxide: carbon dioxide

Line 493, Eucalypt spp. Have: Eucalyptus spp. have

Line 501, Eucalypt spp. Have: Eucalyptus spp. have

Line 500-510, a very long sentence, it should be reorganized.

Line 506-507, Plant Vitrification Solution (PVS) and Preservation of Axillary buds: plant vitrification solution (PVS) and preservation of axillary buds

Line 512, Half: half

Line 515, the full genus name for C. toleriana should be provided.

Line 515, and: and (not italicized)

Line516, Full: full

Line 521, Eucalypt spp. : Eucalyptus spp.

Line 524, 529, Eucalypts: Eucalyptus

Line 528, 538, Eucalypt: Eucalyptus

Line 530, Eucalypt spp: Eucalyptus spp.

Line 532, ttype: type

Line 536, All: all

References (line 551-884): The format of all of the references needs to be revised. Please pay more attentions to the format of author names, paper titles, plant scientific names, journal titles (full name or abbreviation). There is only one word in [50]. [11] is a copy of [10], and [56] is a copy of [55].

There are a lot of minor errors in English language writing, and it is necessary to  revise these errors.

Author Response

Thank you Sir/ Ma’am for your suggestions, considering your valuable feedback report, we have updated the manuscript as follows. We have corrected the manuscript and highlighted in yellow.

Reviewer 3

(1) In ‘2.2.1 Organogenesis’, the authors indicated organogenesis refers to the process of shoot regeneration. Actually, organogenesis should include the regeneration of roots, and this part should be described in the manuscript. Corrected

(2) In ‘2.2.2 Somatic embryogenesis’, the authors indicated it is a pathway for shoot regeneration. Somatic embryogenesis is also the pathway for root or plantlet regeneration. Corrected

(3) Line 253, the authors proposed that somatic embryos may directly develop from pre-embryonic cells. Considering the cultured explants are usually leaves, stems and roots, the sources of pre-embryonic cells should be described. Corrected

(4) Figure 2 provided little information, and the items are repeated, for example, ‘Carbohydrates’ is also ‘Organic and inorganic supplements’ and ‘Culture media’, while the above contents have been described in the text. This figure could be deleted. Corrected

(5) As to ‘4. in-vitro germplasm preservation’, there is only one sub-title (4.1 cryopreservation and cold storage) in this part, and this sub-title could be deleted. On the other hand, the authors should pay more attention to the writing format of all of the topic titles in the manuscript. Corrected

(6) Table 1 and Table 2 should be in the form of three-line table, and the table title should be put above the table. The abbreviations could be put under the table. Corrected

Minor concerns:

Line 25, High-quality: high-quality. Corrected

Line 3, 28, 30, EucalyptEucalyptus. Corrected

Line 46, 68, 81, Eucalyptus: Eucalyptus (italicized). Corrected

Line 67, Traditional and Modern: traditional and modern (not capitalized word). Corrected

Line 86, 87, Tropical and Subtropical: tropical and subtropical (not capitalized word). Corrected

Line 90, and: and (not italicized). Corrected

Line 130, Table 2, Line 156, Line 169, Eucalypt: Eucalyptus. Corrected

Line 155, Additives: additives (not capitalized word). Corrected

Line 161, Five: five. Corrected

Line 164, 165, stage: Stage. Corrected

Line 185, Dormant Axillary: dormant axillary. Corrected

Line 193-207, it is not suitable to capitalize the name of chemical reagents. Corrected

Line 204, the full genus name for C. citridora should be provided. Corrected

Line 213, Nodal: nodal. Corrected

Line 242, 244, 264, 267, Eucalypt: Eucalyptus. Corrected

Line 273, Semisolid: semisolid. Corrected

Line 284, Red: red. Corrected

Line 320, Eucalypt: Eucalyptus. Corrected

Line 329, Tissues of Eucalyptus: tissues of Eucalyptus plants. Corrected

Line 341, Auxin: auxin. Corrected

Line 346, Comparing: comparing. Corrected

Line 362, Gelling: gelling. Corrected

Line 367: Eucalypt: Eucalyptus. Corrected

Line 379, eucalyptus: Eucalyptus. Corrected

Line 382, Eucalypts: Eucalyptus. Corrected

Line 384, Eucalypt: Eucalyptus. Corrected

Line 390-419, 440, 442, it is not suitable to capitalize the name of elements and chemical reagents. Corrected

Line 420, Eucalypt: Eucalyptus. Corrected

Line 433, to promote. The: to promote the. Corrected

Line 434, Eucalyptus: Eucalyptus (italicized). Corrected

Line 454, 455, Use of Red/ Blue LEDs and Florescent: use of red/ blue LEDs and florescent. Corrected

Line 456, Eucalypt: Eucalyptus. Corrected

Line 460, White: white. Corrected

Line 466, Carbondioxide: carbon dioxide. Corrected

Line 493, Eucalypt spp. Have: Eucalyptus spp. have. Corrected

Line 501, Eucalypt spp. Have: Eucalyptus spp. have. Corrected

Line 500-510, a very long sentence, it should be reorganized. Corrected

Line 506-507, Plant Vitrification Solution (PVS) and Preservation of Axillary buds: plant vitrification solution (PVS) and preservation of axillary buds. Corrected

Line 512, Half: half.Corrected

Line 515, the full genus name for C. toleriana should be provided. Corrected

Line 515, and: and (not italicized).Corrected

Line516, Full: full.Corrected

Line 521, Eucalypt spp. : Eucalyptus spp. Corrected

Line 524, 529, EucalyptsEucalyptus. Corrected

Line 528, 538, EucalyptEucalyptus. Corrected

Line 530, Eucalypt spp: Eucalyptus spp. Corrected

Line 532, ttype: type. Corrected

Line 536, All: all. Corrected

References (line 551-884): The format of all of the references needs to be revised. Please pay more attentions to the format of author names, paper titles, plant scientific names, journal titles (full name or abbreviation). There is only one word in [50]. [11] is a copy of [10], and [56] is a copy of [55]. Corrected

Round 2

Reviewer 1 Report

Author should add Limitation, challenges and future direction should be added in separate headings before conclusion 

minor error need to be corrected

Author Response

Comment: Author should add Limitation, challenges and future direction should be added in separate headings before conclusion.

Response: Dear Sir many thanks for your comment. We have modified the manuscript as suggested.

Reviewer 3 Report

i. Table 1 and Table 2 should be in the form of three-line tables.

ii. One more 'Figure 3' was showed in line 188.

iii. The 'spp.' in line 515, 521 should  not be italic.

iv. References (line 574-900): The format of the references needs to be revised. Please pay more attentions to the format of author names, paper titles, journal titles (full name or abbreviation). The editors' name and publisher should be indicated in [51].

The quality of English language writing needs to be improved further.

Author Response

Reviewer2

i. Table 1 and Table 2 should be in the form of three-line tables. 

Response: Corrected.

ii. One more 'Figure 3' was showed in line 188. 

Response: Corrected.

iii. The 'spp.' in line 515, 521 should  not be italic.

Response: Corrected.

iv. References (line 574-900): The format of the references needs to be revised. Please pay more attentions to the format of author names, paper titles, journal titles (full name or abbreviation). The editors' name and publisher should be indicated in [51].

Response: Revised and corrected using zotero.